# Shapley Oracle Pruning for Convolutional Neural Networks

## Abstract

The recent hardware and algorithmic developments leverage convolutional neural networks to considerable sizes. The performance of neural networks relies then on the interplay of even larger pool of, possibly correlated and redundant, parameters, huddled in convolutional channels or residual blocks. To this end, we propose a game-theoretic approach based on the Shapley value, which, accounting for neuron synergies, computes the average contribution of a neuron. A significant feature of the method is that it incorporates oracle pruning, the ideal configuration of a compressed network, to build a unique ranking of nodes that satisfy a range of normative criteria. The ranking enables to select top parameters in the network and remove trailing ones, thus creating a smaller and better interpretable model. As applying the Shapley value to numerous neurons is computationally challenging, we introduce three tractable approximations to handle large models and provide pruning in a reasonable time. The experiments show that the proposed normative ranking and its approximations show practical results, obtaining state-of-the-art network compression. The code is available at https://anonymous.4open.science/r/shapley_oracle_pruning1/.

## 1 Introduction

Convolutional neural network models are powerful function approximators that have proved their efficacy by achieving state-of-the-art results for various computer vision tasks (58; 14; 18; 64; 10; 19). Their performance results from taking advantage of a hierarchical structure of heavily parametrized layers which grew in particular with the advent of such networks as AlexNet (24), VGG (55) and ResNet (14). Thus, the startling capabilities come at the cost of large models, slower inference speed and a need for specialized processing units.

At the same time as the models grow larger, incorporate more convolutional channels and introduce alterations in form of skip connections, the knowledge about the network parameters on the microscopic level remains largely unexplored. In this work, we consider the individual neurons and their role in the larger scheme of the model performance, thus taking a local and bottom-up approach. Understanding the role of each neuron is useful to confidently select the important parameters and eliminate neurons of less significance, resulting in a smaller neural network size (48; 47; 36; 31).

The current literature (47; 48; 28) proposes a range of approaches that assess the role of a neuron by its sensitivity to the loss function, measured by its derivative with respect to the parameters. Those with decreasing sensitivity can be thrown away. Nonetheless, the main issue with these approaches is that the parameters are selected greedily and treated independently of each other. This may result in removing neurons that in cooperation with others are useful for the network performance.

In contrast, this paper proposes to look at the role of a neuron while taking account of network synergies, that is collections of parameters of arbitrary size, and subsequently measure the individual record in relation to its group performance. To this end, we employ the concept of the Shapley value (54) from coalitional game theory, which looks at the marginal contribution of a node to its coalition. We show how this elegant concept works on modern architectures and produces realistic speed-ups due to a structured approach to pruning. In summary, the method:

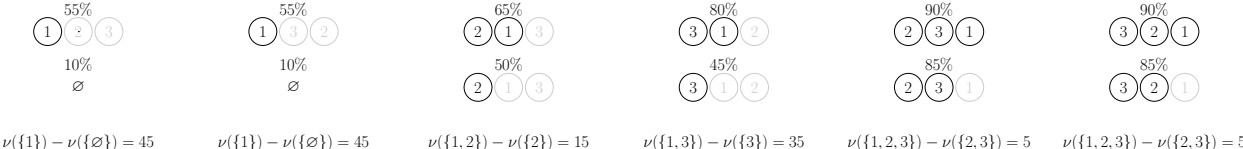

Figure 1: An example of computing the Shapley value of node 1, $\varphi_1$, in a neural network according to the definition from Eq. (4). We consider a single layer with three nodes (numbered 1,2 and 3) and compute the marginal contribution of node 1 in each of the 3! permutations of all the nodes. The bold nodes represent coalitions. A coalition is formed by appending nodes from left to right. The upper row includes the coalitions with node 1, the lower row contains the corresponding coalitions without node 1. The average contribution is then $\varphi_1 = \frac{45+45+15+35+5+5}{3!} = \frac{150}{6} = 25$. The percentage illustrates the characteristic function, that is the accuracy of the network containing only coalition nodes. The accuracy of the full network is 90% and with all the nodes removed 10%. By performing similar computations, we can calculate that $\varphi_2 = 25$, $\varphi_3 = 30.3$. This indicates that node 3 on average contributes the most to the network, and according to the Shapley Oracle pruning would be the most important node in the network.

- Incorporates the game-theoretical concept of the Shapley value which precisely quantifies the impact of a neuron (convolutional channel/3D filter or linear layer unit) on the model performance by means of computing its average marginal contribution.

- Neatly utilizes the concept of oracle pruning, the optimal pruning configuration, to create a normative ranking of units.

- Generalizes and systematizes current approaches through three approximation schemes: partial $k$-greedy, random permutation sampling, and weighted least-squares regression. These approximations render the Shapley value approach applicable to larger parameter spaces and, contrasting to many pruning approaches, provide fast compression.

- Applies structured pruning and provides heavily compressed networks that can be readily applied for vision tasks.

## 2   Related work

This work aims to propose a way to compress the network via quantifying the overall impact of its parameters on network performance, and therefore it relates to the compression literature that measures the parameter utility by its sensitivity to the changes in the loss function. In fact, measuring parameter sensitivity was one of the main initial attempts at network pruning in the late 80s and 90s. In particular, (7) sums the incremental changes to the weights during the course of training.

**Derivative pruning.**   Generally, in order to measure the change in the loss function due to removing a parameter $p$, we aim to compute the derivative $\partial \mathcal{L}/\partial p$. This derivative has been locally approximated with a Taylor series, and then further simplified by only considering the second order term (28; 13). Due to difficulties in computing the Hessian, further approaches aim to approximate it with the empirical estimate of the Fisher information (60) or abandon it altogether and focus on using a first-order approximation (48), which uses the variance of the first order, $p\partial \mathcal{L}/\partial p$, rather than the first order itself (which tends to 0 for a well-trained model).

The above line of work belongs to the greedy approaches, which measure the impact on single parameters or channels and remove those which contribute little to the overall performance of the model. This work is meant to look beyond. The Shapley value is a measure that computes the marginal contribution over all subsets of parameters. Depending on the application one may want to compress the network to an arbitrary size $K$, therefore the proposed approach, where we look at subsets of each size $K$, is a more sensible approach.

As mentioned in the previous paragraph, one may look at the role of different units in the network. While the earlier pruning literature deals with individual parameters (also within convolutional layers), coined as unstructured pruning, in recent years we have seen more focus on structured pruning (5; 30; 61). The latter two algorithms among the second-derivative methods (48; 60) are also examples of structured pruning, and earlier work has been also adapted (25). In a similar fashion, (29) proposes the Shapley value to prune multi-layer perceptrons in an unstructured manner. On the other hand, we advocate the Shapley value as a viable way to compress convolutional channels within modern convolutional neural networks, including residual blocks. The such approach offers practical solutions e.g. VGG models are reduced from 79 MB to merely 3,4MB by using the proposed method.

**Matrix decomposition.** The idea of exploring the structure of the network, albeit in a different form, motivates the line of work that proposes to decompose the weight matrices to create low-rank representation of the network parameters (53; 50; 32), i.e. $\mathbf{W} \in \mathbb{R}^{u \times v}$ is decomposed as $\mathbf{W} = \mathbf{AB}$ where $A \in \mathbb{R}^{u \times k}$ an $B \in \mathbb{R}^{k \times v}$. Much of this line of work uses Singular Value Decomposition (SVD) (62; 9; 22). For orthogonal matrices $U \in \mathbb{R}^{v \times k}$, $V \in \mathbb{R}^{v \times k}$ and a diagonal matrix of singular values, $\Sigma \in \mathbb{R}^{k \times k}$, we can express A as $W = U\Sigma V^T$. The diagonal elements of $\Sigma$ sorted in decreasing order can be used to approximate $W$. The idea of determining the rank of the matrix through SVD is loosely related to extracting the uncorrelated "base" features of the network. However, in the decomposition methods, the interpretability of results is limited and current approaches rather focus on computational savings through sparser representations of the network.

**The Shapley value.** The Shapley value has been proposed in the mid-20th century (54) and is arguably the most important concept in coalitional game theory, the branch that deals with groups of individuals rather than single players. Although the original idea behind the Shapley value is to find a fair division pay-off, a corollary to this idea is to be able to find a reliable way to learn the most important players in a group, a notion known as centrality (46). This is the idea that we explore in this work.

The application of the Shapley value in machine learning has been recent (39; 57), which could be attributed to its computational difficulty. Different ways to approximate the Shapley value have been proposed. Random sampling has many similar variants (1; 6; 4), which we also adopt in this work. However, the widespread adoption of the Shapley value is due to the reinterpretation of the Shapley value as a linear function (8; 43). The approach has been widely discussed (21; 2). It is adapted in our third approximation scheme. Recently, concurrently with our work, (12) proposes a multi-armed bandit algorithm where they approximate Shapley values via random permutations based on confidence intervals. Moreover, while (12) underscores the interpretability and provides visualizations as well as some applications to fairness assessment and finding filters vulnerable for adversaries adversaries, in this work, we propose the concept as a viable solution for network compression. The proposed three approximation schemes can be applied to medium-to-large parameter spaces. They can be adjusted appropriately given the available computational budget.

## 3 Problem formulation

To compress a network, we measure the impact of particular parameters of a neural network model on its end performance and provide a rank of the parameters in terms of their importance to the network's outcome. Although the proposed framework is general for any type of parameter units, in this work we concentrate on network channels (or 3D filters) in convolutional layers and single weights in fully-connected layers. We will refer to these units collectively as nodes or neurons. In residual blocks, we enforce the same input and output to the block.

Let $\mathcal{N}^l$ describe the set of nodes in a layer $l$. For clarity we will typically omit the superscript and consider the set of nodes $\mathcal{N}$ to be the set within a single layer, and $|\mathcal{N}| = N$ to be the number of nodes in a single layer. Then let $\mathcal{K} \subseteq \mathcal{N}$ be a subset of nodes, and $|\mathcal{K}| = K$. Let $n_i \in \mathcal{N}$ be a single node and $i$ be the index of a particular node, i.e. $i \in [1, N]$. Subsequently, let $\nu : \mathcal{K} \subseteq \mathcal{N} \to \mathbb{R}$ be a function which takes the subset of nodes $\mathcal{K}$ as the input and outputs a number which is a single value, a reward (or cost) assessing the subset, for instance, the accuracy of the pruned network on the validation set. In the next section we

Table 1: Comparison between terminologies in game theory (left) and our formulation of CNN rankings (right).

| Game theory | Ranking in CNNs | Notatian |
|---|---|---|
| player | node (or neuron) (conv or fully-connected) | $n$ |
| characteristic function | accuracy on validation set | $\nu$ |
| coalition | subset of neurons | $\mathcal{K}$ |
| grand coalition | set of original neurons in a layer | $\mathcal{N}$ |
| coalition cost | accuracy loss after removing a subset of neurons | $\nu(\mathcal{K})$ |

will reformulate the above terms to incorporate the game-theoretical framework to quantify the impact of each node interactions of network parameters.

## 4 Game theoretical neuron ranking

This section addresses the problem of quantifying the contribution of a single network node to the overall network performance. Assume a group of nodes work together which results in a gain because of their cooperation. However, each node may have a different role and contribution in this task. How do you fairly distribute the total gain to individual players? This is for the problem of network compression, when building a smaller network, we aim to fairly attribute the performance gain to individual nodes. In particular, we phrase this problem as a game and employ the concept from the coalitional game theory, called the Shapley value, which precisely quantifies the neuron's importance as its average marginal contribution for the network's performance. Given this neuron's importance in form of the Shapley value, we may then sort the values and remove the nodes with the smallest Shapley value, that is those that least contribute to the network performance.

### 4.1 Coalitional game theory

Let a node, as defined in the previous section, be called a player (we will use player/node interchangeably), the set of all players $\mathcal{N} := \{0, \ldots, N\}$ the *grand coalition* and a subset of nodes $\mathcal{K} \subseteq \mathcal{N}$ a coalition of players. Subsequently, we assess the utility of a given coalition, i.e. of a given subset of nodes. To assess quantitatively the performance of a group of players, each coalition is assigned a real number, which is interpreted as a payoff or a cost that a coalition receives or incurs *collectively*. The value of a coalition is given by a *characteristic function* (a set function) $\nu$, which assigns a real number to a set of players. A characteristic function $\nu$ as defined before maps each coalition (subset) $\mathcal{K} \subseteq \mathcal{N}$ to a real number $\nu(\mathcal{K})$. Therefore, a coalition game is defined by a tuple $(\mathcal{N}, \nu)$.

In our case, a subset would be a set of nodes in the network (in a single layer), and the characteristic function evaluates the performance, such as accuracy, of the network when we only keep the set of nodes and prune the rest from the network. This is, the output of the characteristic function applied to the grand coalition (denoted by $\nu(\mathcal{N})$) is the performance of the original network with all nodes in place, while applied to a subset (denoted by $\nu(\mathcal{K})$) resembles the performance after removing the nodes that are missing in the subset. The choice of the characteristic function is a critical component of a coalition game, since it defines the quantity of interest. We focus on accuracy as characteristic function in this work, and formally define the function as

$$\nu(\mathcal{K}) := \frac{\sum_{i=0}^{M} \delta_{\hat{y}_{\mathcal{K}}^i, y^i}}{M}, \tag{1}$$

where $M$ is the number of data samples for the evaluation, $\hat{y}_{\mathcal{K}}^i$ the output of the neural network for the $i$-th input when we only consider nodes in $\mathcal{K}$, $y^i$ is the corresponding ground truth label, and $\delta_{\hat{y}_{\mathcal{K}}^i, y^i}$ is the

Kronecker-Delta,

$$\delta_{\hat{y}^i_{\mathcal{K}}, y^i} = \begin{cases} 1, & \hat{y}^i_{\mathcal{K}} = y^i \\ 0, & \text{otherwise} \end{cases} . \tag{2}$$

Based on this definition, the Shapley values therefore estimates the contribution to the quantity

$$\nu(\mathcal{N}) - \nu(\emptyset) = \frac{\sum_{i=0}^{M} \left( \delta_{\hat{y}^i_{\mathcal{N}}, y^i} - \delta_{\hat{y}^i_{\emptyset}, y^i} \right)}{M},$$

where $y^i_{\emptyset}$ indicates the baseline output when we remove all nodes $\mathcal{N}$ from the graph. Note that this could even become negative if the removal of certain nodes lead to a structured mis-classification of inputs. While in our case we define this characteristic function as the accuracy of the network, it could be simply replaced by another metric such as squared error, log-loss, F1 score, type-1 or type-2 error etc. and, thus, can also be easily formulated for regression problems.

Up until now, we have defined the payoff given to a group of nodes. The question now remains how to assess the importance of a single node given the information about the payoffs for each subset of nodes. To this end, we employ the concept of the Shapley value about the normative payoff of the total reward or cost.

## 4.2   Shapley value

In this section, we compute the contribution of a single node (or neuron) to the network's performance. This contribution is given by a number, the Shapley value. The concept introduced by Shapley ([54]) is a division payoff scheme which splits the total payoff into individual payoffs given to each separate player. These individual payoffs are then called the Shapley values. The Shapley value of a player $i \in \mathcal{N}$ is given by

$$\varphi_i(\nu) := \sum_{\mathcal{K} \subseteq \mathcal{N} \setminus \{i\}} \frac{1}{N \binom{N-1}{|\mathcal{K}|}} (\nu(\mathcal{K} \cup \{i\}) - \nu(\mathcal{K})). \tag{3}$$

Intuitively, we look at a subset of size $K$ and compute the difference a node $i$ makes to the network's performance if we add that node to the subset. Then we repeat it for all the subsets and compute the average. This intuition is further illustrated in Fig. 1. The value $\varphi_i(\nu)$ quantifies the (average) contribution of the $i$-th player to a target quantity defined by $\nu(\mathcal{N}) - \nu(\emptyset)$, that is the output of the characteristic function applied to the grand coalition minus the output when applied to the empty set. The sum over the Shapley values of all nodes is equal to this target quantity, $\nu(\mathcal{N}) - \nu(\emptyset) = \sum_{i=0}^{N} \varphi_i(\nu)$. In our case, the grand coalition is the original, non-compressed network and the empty set is the network with random performance (which represents a compressed network where all the units, that is a whole layer, are removed). Using the Shapley value scheme ensures that the contributions are estimated in a 'fair' way, that is according to a mathematically rigorous division scheme that has been proposed as the only measure that satisfies four normative criteria regarding the fair payoff distribution. We describe these criteria in detail in the Supplement.

The Shapley values is computed based on the the average marginal contribution of a node. To provide some further intuition, we describe the process of a finding a single marginal contribution. This can be done with the process of building a coalition via a permutation of nodes. When a coalition has no members (empty set), and the neuron $n_1$ joins the coalition, the value of its marginal contribution is equal to the value of the one-member coalition, $\nu(\{n_1\}) - \nu(\emptyset) = \nu(\{n_1\})$ where $\{n_1\} = \mathcal{K}$. Subsequently, when another player $n_2$ joins this coalition, its marginal contribution is equal to $\nu(\{n_1, n_2\}) - \nu(\{n_1\})$. The process continues until all the nodes join the coalition (also see Fig. 1). The order of nodes, which builds subsequent coalitions to finally form the grand coalition, can be represented as a permutation of nodes, e.g. $n_1 n_2 ... n_{N-1} n_N$ or $n_5 n_3 n_7 ... n_N ... n_2$. One permutation represents one way to form a coalition. There are $N!$ permutations of $N$ nodes, meaning that there are $N!$ different ways to form a coalition. Seeing this exponential complexity to evaluate Eq. (3), we will discuss some more practical approaches to approximate the Shapley values in the following Section.

### 4.3 Three approximations of the Shapley value

Although the Shapley value is theoretically an elegant concept, the computational complexity of the original form hinders its applicability. In the case of measuring the importance of a node, this has twofold impact. On the one hand, the sheer number of operations for a layer with larger number of nodes is prohibitive (e.g. a layer 64 channels requires $10^{89}$ operations). On the other hand, for a given coalition, the constant cost to compute the output of the characteristic function is a forward pass of a neural network which in itself is significant. As a result, we propose three distinct ways to reduce the computational burden and obtain a meaningful approximation of the Shapley value. The first scheme reduces the size of the coalition, the second one performs permutation sampling and the third reformulates computing the Shapley value into a weighted linear regression problem. We compare all three schemes in the ablation study in the experiments section.

#### 4.3.1 Approximation via partial Shapley Value

The common scheme to assess the importance of a node is to remove the node ("leave-one-out") and check the impact of its absence on the network evaluation metric (47; 48). This case turns out to be a special case of the Shapley value when the size of a coalition is restricted to 1. This approach may be efficient but can be problematic, intuitively, because the loss we incur by removing two nodes together is not the same as the sum of losses incurred by removing them individually. This case is illustrated in the example of Fig. 1 where greedily removing two nodes (1 and 2) would lead to suboptimal outcome. We elaborate on this issue in the Supplement.

The Shapley value naturally generalizes the idea of greedy deletion to assessing the importance of a node when considering its removal as a part of group of nodes (which is the case in the network compression). In the original formulation in Eq. 3, the size of the subset is any $K \in [1, N]$. In the "partial" Shapley value we restrict the size of the subset $\mathcal{K}$. The greedy approach considers the coalition size $K = N - 1$. We subsequently extend the approach to $K < N - 1$, that is compute the losses incurred for every pair of nodes, triple, etc. The number of computations is given by $\binom{N}{K}$ which gets larger as $K$ approaches $N/2$. Thus due to computational complexity, we also restrict $K \geq M_1$ such that $M_1 \gg N/2$ ($M_1$ is much bigger than $N/2$). Because $\binom{N}{K} = \binom{N-K}{K}$ we may also include the small $K < M_2$ and similarly $M_2 \ll N/2$.

Notice that the computational burden is reduced from exponential time complexity $O(2^{n-1})$ to polynomial time $O(n^k)$. The reduced sample is a promising direction but for large $k$ it may still be expensive. We may resort then to a more flexible sampling scheme which we subsequently present.

#### 4.3.2 Approximation via averaging permutations

An alternative, yet simple and efficient, way to approximate Shapley values is by averaging over a limited number of random permutations (see Sec. 4.2 for the definition of permutations.). Generally, we can rewrite Eq. (3) as

$$\varphi_i(\nu) = \sum_{\pi \in \Pi(N)} \frac{1}{N!} (\nu(\pi_K \cup \{i\}) - \nu(\pi_K)), \tag{4}$$

where $\pi \in \Pi(N)$ represents one permutation of the players. This is, averaged over all possible permutations, we obtain the Shapley value for the $i$-th player based on Eq. (4). While evaluating $N!$ many permutations is intractable, this formulation allows to have an unbiased approximation of the Shapley values when we limit the number of permutations we evaluate. This is, if $\pi$ is uniformly sampled from $\Pi(N)$ with probability $\frac{1}{N!}$, (4) converges to the true $\varphi_i(\nu)$ (if normalized by the number of sampled permutations accordingly). For further insights regarding the statistical properties of this approach, we refer the reader to (57).

Practically, we only need to draw a few permutations $\pi$ in order to obtain a reasonably good approximation of the Shapley values. This is, given a permutation $\pi$, we can simply iterate through the order of nodes defined by $\pi$ and evaluate the characteristic functions. We can then obtain an approximation $\hat{\varphi}_i(\nu)$ by

$$\hat{\varphi}_i(\nu) = \frac{1}{S} \sum_j (\nu(\{i\} \cup \bar{\pi}_j(i)) - \nu(\bar{\pi}_j(i))),$$

where $S$ is the number of randomly drawn permutations and $\bar{\pi}_j(i)$ indicates the set of players before the $i$-th player in the $j$-th permutation.

This approximation technique has the advantage that it is quite robust in practice and provides more flexibility as we can adjust the number of samples constrained by our computational budget. Unlike the previous scheme, which requires the computation of $\binom{N}{K}$ to obtain the same number of samples (e.g. test set size) for each channel, in this scheme we may adjust the number of iterations and perform $N$ validation tests to obtain samples for each channel. It would also be possible to introduce an early stopping criterion in case the Shapley values do not change significantly anymore between the evaluation of two (or more) permutations. Nevertheless, if $N$ is large, even one data sample that requires $N$ forward passes may be costly. The reinterpretation of the Shapley value as a regression problem can provide the approximation even with a few forward passes.

### 4.3.3 Approximation via weighted least-squares regression

The third Shapley value approximation is somewhat different to the previous two as it describes the sets as binary vectors with the vector dimensionality equal to the total number of players. This binary vector then indicates whether a player is present in a subset or not. This allows to formulate (3) as a weighted least-squares regression problem.

Nevertheless since the exact Shapley value could only be obtained with the exponential number of binary vectors, we again resort to sampling. Thus, a sample in form of a permutation $\pi$ is drawn from the set of $N!$ permutation in a similar fashion as in the previous section. However, here we do not compute the marginal contributions for all the players (although that could be possible). Instead, we rather sample subsets and its characteristic function value, i.e. we sample values $\nu(\mathcal{K})$ where $\mathcal{K}$ is based on $\pi$. Given the subset $\mathcal{K}$, we create a binary vector $\mathbf{v}$ s.t. $|\mathbf{v}| = N$, $\nu(\mathbf{v}) = \nu(\mathcal{K})$ and

$$\mathbf{v}_i = \begin{cases} 1 & \text{if } i \in \mathcal{K}, \\ 0 & \text{otherwise.} \end{cases}$$

Alternatively, we can also sample the binary vectors directly by assigning $1/2$ probability to be either 0 or 1 to each vector entry or sample the vector based on the probability $1/\binom{N}{K}$ for a subset of length $K$.

Consider then the Shapley values $(\varphi_0(\nu), \ldots, \varphi_N(\nu))$ to be the weights of the binary vector $\mathbf{v}$. As stated by (8), a formulation in this form then allows to obtain the Shapley values as the solution of

$$\min_{\varphi_0(\nu),\ldots,\varphi_N(\nu)} \sum_{\mathcal{K} \subseteq \mathcal{N}} \left[ \nu(\mathcal{K}) - \sum_{j \in \mathcal{K}} \varphi_j(\nu) \right]^2 k(\mathcal{N}, \mathcal{K}), \tag{5}$$

where $k(\mathcal{N}, \mathcal{K})$ are so called Shapley kernel weights. These are defined as:

$$k(\mathcal{N}, \mathcal{K}) := \frac{(|\mathcal{N}| - 1)}{\binom{|\mathcal{N}|}{|\mathcal{K}|} |\mathcal{K}| (|\mathcal{N}| - |\mathcal{K}|)},$$

where $k(\mathcal{N}, \mathcal{N})$ is set to a large number due to the division by 0. In practice, Eq. (5) can then be solved by solving a weighted least-squares regression problem with the solution

$$\phi = (\mathbf{V}^T \mathbf{K} \mathbf{V})^{-1} \mathbf{V} \nu,$$

where $\mathbf{V}$ is a matrix consisting of the above defined binary vectors, $\mathbf{K}$ the Shapley kernel weight matrix, and $\nu$ a vector with the outcomes of the characteristic function applied to the corresponding subset in $\mathbf{V}$.

As mentioned before, it is possible to approximate (5) by only sampling a few binary vectors (according to their probability). Comparing this approach with the approach described in Section 4.3.2, both have the same convergence rate, while the weighted least-square solution allows to incorporate regularization terms such as $L_1$ to prune small Shapley values, but suffer from some numerical instabilities in practice. Further, introducing an early-stopping criteria for the weighted least-squares solution is less straightforward than for the permutation based approach.

Table 2: The ablation study of the Shapley value approximation schemes. As a metric, we use the Jaccaard index, $J(\mathcal{K}_O, \mathcal{K}_R) := |\mathcal{K}_O \cap \mathcal{K}_R| / |\mathcal{K}_O \cup \mathcal{R}_R|$, where $\mathcal{K}_O$ is an oracle subset and $\mathcal{K}_R$ is a subset provided by a given ranking. We measure the overlap between the subsets of size $K$ (where $K \in [1, 5]$) and sum the Jaccard indices weighted by the size of the subset. The experiment is performed for two convolutional layers with 10 and 20 nodes. The Shapley value approximation methods we utilize here are two variants of the partial approximation (Leave-one-out (Partial-1) and Partial-3), weighted linear regression and permutation sampling. We also include the true Shapley value, the value we are try to approximate, and the ground truth Oracle ranking which is computed using the oracle optimal subsets (note the distinction between the Oracle ranking and the oracle subsets). The upper table presents the results for selecting top $K$ nodes to build a layer, and the bottom table the best nodes to remove from the layer.

| Rank | Leave-one-out | Partial-3 | Regression | Permutations | SV ($\varphi$) | Oracle |
|---|---|---|---|---|---|---|
| | | | | Best to keep | | |
| $N = 10$ | 0.702 | 0.584 | 0.678 | 0.676 | 0.688 | 0.874 |
| $N = 20$ | 0.332 | 0.332 | 0.398 | 0.452 | - | 0.6 |
| | | | | Best to remove | | |
| $N = 10$ | 0.916 | 0.878 | 0.916 | 0.918 | 0.944 | 1 |
| $N = 20$ | 0.33 | 0.364 | 0.33 | 0.372 | - | 0.74 |

## 5 Experiments

In this section we present the experiments to investigate the Shapley value algorithms for building node importance ranking and compressed network architectures. Firstly, we perform ablation study where we compare the performance of the proposed approximation algorithms with the ground-truth Shapley value in a small-scale experiment, and for comparison introduce, so called optimal ranking. Secondly, we showcase the network compression results on both small and large-scale architectures.

### 5.1 Ablation study

A good ranking of the top contributing nodes is the key aspect in network pruning. The proposed approach by means of the Shapley value is supposed to produce such a reliable rank. In this study we verify different ranks produced by the Shapley value and its approximations and compare it with the 'oracle' rank which is produced by the *oracle*, the optimal pruning configuration. We introduce the notion of the oracle ranking, which is the rank created based on the oracle subsets, to upper bound the number of correct top-$K$ nodes from a ranking. We define an *oracle subset* of size $K$ as an optimal in the following way. In a task to prune $K$ out of $N$ nodes in a layer, the oracle 'knows' the network accuracy for every subset of size $K$ removed [1] (out of $\binom{N}{K}$ subsets), and it selects a subset which results in the smallest accuracy loss, which is the oracle subset. We also assume that a priori we do not know the user's intention of how much one wants to prune the network, that is $K$ is unknown, which makes the problem computationally hard.

Oracle ranking is a fixed ordering of nodes where top-$K$ positions in a ranking are selected for our task (that is, top-$K$ would be pruned). Oracle ranking is made with $\binom{N}{K}$ subsets in a way to maximize the number of nodes in a ranking overlapping with the nodes in oracle sets. Let us note that the ranking would be different if we used $M < N$ and $\binom{M}{K}$. That is, the oracle subset of size $K$ and $K + 1$ may contain different elements, which prevents creating one ranking where the top $K$ positions contain elements from the oracle $K$-size subset and top $K + 1$ positions contain elements from the $(K + 1)$-size oracle subset. For example, consider a hypothetical scenario where we want to remove only one or two nodes, and consider the following oracle subsets. The least useful node is $\{5\}$ and the least useful two-element subset is $\{2, 7\}$. Then neither of possible reasonable ranks ($[5, 2], [5, 7], [2, 7], [7, 2]$, top position from left to right) can select both one-element

---

[1]That is the size of the subset we keep is $N - K$. We slightly abuse the wording and, when indicated, refer to subset of size $K$ as the size of the subset that we remove.

and two-element set which would coincide with the oracle sets). Nonetheless despite this rigid nature of a rank, in this ablation study we argue that a subset constructed based on the top-$K$ rankings is reasonably *close* to the ground truth oracle subset of size $K$, and provide a flexible scheme where we assume not to know $K$ beforehand. The ablation study evaluates how close these two subsets are by evaluating how many elements (unordered) coincide. As argued above, no ranking guarantees to reconstruct the oracle subsets. Yet we build here a ranking based on the oracle subsets which maximizes the intersection between a ranking and the subsets. The ranking is supposed to upper-bound the performance of any ranking, including the Shapley value-based rankings. The details of how we create this Oracle ranking can be find in the Supplement.

In addition to the Oracle ranking, in this ablation study we include the ranking produced by the exact Shapley value as defined in Sec. 4.2 and the four rankings are approximation schemes of the Shapley value, that is a partial Shapley with $k = 1$ (greedy/leave-one-out ranking) and $k = 3$, random sampling based on permutations and weighted linear regression approximation. The experiment is performed on the first and second layer of the reduced Lenet-5 with 10 and 20 nodes, respectively. In the case of the first layer, relatively small scale of the layers allows for the computation of all the $\binom{N}{K}$ subsets and therefore the exact computation of the Shapley value according to the Eq. (3) is possible. Thus, we are able to check how well the approximation algorithms perform in comparison to the exact Shapley value. In the second layer, as in most of the cases of large network layers, we must resort to the approximation schemes.

The results of the ablation study are summarized in Table 2. We distinguish two cases of the problem, the first one is to find top-$K$ nodes which constitute the optimal set for a layer of size $K$. The second case describes finding the set of the most redundant nodes which can be removed from the layer with the least loss in the predictive performance. In other words, in the first case we end up with a layer of size $K$ while in the second one with the layer of size $N - K$.

The results bring a number of interesting findings. The classical Shapley value in its exact form finds the ranking which resembles most the Oracle ranking. The case of the leave-one-out (Partial-1) ranking is particular. In the case of $K = 10$, the approach obtains excellent results mirroring the oracle and Shapley results. However, its performance drastically deteriorates for $K = 20$. We surmise, that this simple approach works well for smaller layers where there is less redundancy and correlation between weights, and the impact of removing a single node resembles its overall role in the network. However, when the number of nodes increases, there is more interplay and correlations between parameters which can be only discerned by looking at the subsets of higher order. As a result, we see leave-one-out as a useful method which given limited computational resources can be a viable alternative. However, for larger networks, it seems advisable to resort to more complex methods as proposed in this work. Furthermore, the regression and permutation based approximation technique for the Shapley values show a remarkably good performance.

To sum up, the best possible ranking to compute is based on the Shapley value. Then this ranking will be the closest to the oracle pruning. However, as it is most often not possible, both regression and permutation sampling approximations are very good alternatives. And finally, when computational budget is small, leave-one-out approach could be a worthwhile option.

## 5.2 Compression

The experiments are performed on popular benchmarks, MNIST (27), which was trained on the LeNet-5 network (26), and CIFAR-10 (23) trained on the VGG15 (55) and Resnet-56 (14), and Imagenet (24) trained on Resnet-50 (14). For each experiment, we divide the entire dataset into three parts, training, validation and test dataset. Firstly, we train each of the architectures on the training dataset to obtain a pre-trained model. Then for each layer we apply our algorithm to build a ranking of the nodes. We apply the Shapley value method to select the significant channels. The outputs of the characteristic function for the Shapley values are obtained through multiple forward runs of the models on the validation dataset. We then remove the nodes with smallest Shapley value (on average those nodes contribute least to the predictive performance on the network). To prune the parameters, during the node selection procedure we mask the subsets of features (weights and biases, the batch normalization, running mean and variance parameters). Once the best pruned model is selected, we create a thin architecture which is practically smaller and faster than the original model. Once the selected node rankings are obtained, we retrain the pruned network on the entire

| Method | Error | FLOPs | Params |
|--------|-------|-------|--------|
| **Shapley (perm)** | **1.04** | **131K** | **4.5K** |
| Dirichlet (3) | 1.1 | 140K | 5.5K |
| BC-GNJ (41) | 1.0 | 288K | 15K |
| BC-GHS (41) | 1.0 | 159K | 9K |
| RDP (51) | 1.0 | 117K | 16K |
| FDOO (100K) (59) | 1.1 | **113K** | 63K |
| FDOO (200K) (59) | 1.0 | 157K | 76K |
| GL (61) | 1.0 | 211K | 112K |
| GD (56) | 1.1 | 273K | 29K |
| SBP (49) | **0.9** | 226K | 99K |
| Baseline (26) | 0.73 | 2549K | 688K |

Table 3: Comparison of pruned models on **LeNet-5** on MNIST. Top-1 error, computational complexity in terms of FLOPs, and number of parameters are reported.

training set and report the test accuracy in the tables below. The pruned checkpoints are available at our codebase.

| Method | Error | FLOPs | Parameters |
|--------|-------|-------|------------|
| **Shapley (perm)** | **7.91** | **43.0M** | **0.68M** |
| Dirichlet (3) | 8.48 | **38.0M** | 0.84M |
| Hrank (37) | 8.77 | 73.7M | 1.78M |
| BC-GNJ (41) | 8.3 | 142M | 1.0M |
| BC-GHS (41) | 9.0 | 122M | 0.8M |
| RDP (51) | 8.7 | 172M | 3.1M |
| GAL-0.05 (38) | 7.97 | 189.5M | 3.36M |
| SSS (20) | 6.98 | 183.1M | 3.93M |
| VP (65) | 5.72 | 190M | 3.92M |
| Baseline (55) | 5.36 | 313.7M | 15M |

Table 4: Comparison of pruned models on **VGG-16** on CIFAR-10. Notice that the more accurate models are also considerably larger.

We perform the compression of four network architectures. In Lenet and VGG, we prune both convolutional and fully-connected layers. Each layer is assigned a different pruning rate. In Resnet-50 and Resnet-56 we prune the inner convolutional channels of a block, and to maintain the dimensionality of a skip connection, the same channels for the input and output of the residual block or bottleneck module. Removing a channel in a layer consequently results in removing an input channel in the subsequent layer. This means that, for example in the case of Resnet, pruning removes parameters in both layers of each block.

The results of the Shapley pruning are presented in comparison to the recent work in compression literature which include Dirichlet pruning (3), Hrank (37), Bayesian compression (42), GAN-based compression (38), Radial and Directional Posteriors (51), LASSO Regression (17), Scaling factors (20), FLOPs as direct optimization objective (59), Structured Bayesian pruning (49), Structured sparsity (61) and Neuron dissimiliarity (56), ThiNet (45), Group sparsity based compression (33), Differentiable meta pruning (34), Discrimination-aware channel pruning (DCP) (66), Geometric median (16), Adapt-DCP (40), AutoPruner (44), Soft filter pruning (15), ResRep (11). The numerical results come from the respective papers.

We conduct compression by means of the three Shapley-value approximation schemes, and subsequently present the best result among the three approximations. We use 1000-5000 samples for the approximation via least-square regression, and 5-10 samples (each multiplied by the size of the layer) for the approximation using random permutations. Although we use different samples, the samples could be shared between both

| Method | Error | FLOPs | Parameters |
|---|---|---|---|
| **Shapley (regr)** | **8.91** | **13.64M** | **0.24M** |
| Dirichlet (3) | 9.13 | 13.64M | 0.24M |
| Hinge (33) | 7.35 | 30.58M | 0.18M |
| Hrank (37) | 9.28 | 32.53M | 0.27M |
| **Shapley (kern)** | **6.82** | **46.38M** | **0.34M** |
| DHP (34) | 7.06 | 49.78M | 0.35M |
| GAL-0.8 (38) | 9.64 | 49.99M | 0.29M |
| KSE (35) | 7.12 | 50M | 0.36M |
| CP (17) | 9.20 | 62M | - |
| NISP (63) | 6.99 | 81M | 0.49M |
| Baseline (14) | 7.05 | 127.4M | 0.86M |

Table 5: Comparison between pruned **ResNet-56** on CIFAR-10 (left) and the corresponding benchmarks. Top-1 Error and the compressed model parameters and FLOPs are reported.

| Method | Error | FLOPs Ratio | Params Ratio |
|---|---|---|---|
| **Shapley (regr)** | **27.4** | **59.1** | **60.42** |
| SFP (15) | 25.39 | 58.2 | – |
| GAL-0.5 (38) | 28.05 | 56.97 | **83.14** |
| SSS (20) | 28.18 | 56.96 | 61.15 |
| HRank (37) | 25.02 | 56.23 | 63.33 |
| ResRep (11) | **23.85** | 54.54 | - |
| AutoPruner (44) | 25.24 | 48.79 | – |
| Adapt-DCP (40) | 24.85 | 47.59 | 45.01 |
| FPGM (16) | 25.17 | 46.50 | – |
| DCP (66) | 25.05 | 44.50 | 48.44 |
| ThiNet (45) | 27.97 | 44.17 | – |
| Baseline (52) | 23.87 | 0.00 | 0.00 |

Table 6: Comparison between pruned **ResNet-50** on Imagenet (right) and the current methods. Top-1 Error and the compression rates are reported. Higher ratios indicate higher compression and smaller models.

approximation techniques. In three architectures, our proposed method outperforms the current approaches and obtains models which are both faster (in terms of FLOPs) and also smaller (by the number of parameters) given the same computational budget. In Resnet-50, we aim to show higher compression rates preserving competitive accuracy.

Lenet5 results show that we are still able apply more compression to this common benchmark. The resulting network is 17x faster and almost 120x smaller compared to the LeNet-5-Caffe with the original 20-50-800-500 architecture. In the case of both VGG and Resnet56, given similar parameter budget, the Shapley pruning is the only method whose error falls below 8% and 9%, respectively. Noteworthy, both models are 5x faster than some of the recent work. The size of VGG-16 is reduced to 3.4MB and Resnet to 0.35MB.

# 6 Conclusion

To conclude, we presented the Shapley Oracle pruning method which links the network compression concept of oracle pruning with the important solution concept in game theory. As such, we can find reliably important nodes in the network, and given a range of approximation schemes, pruning can be done in a reasonable time. Given its normative desirable criteria, this theory-backed fair ranking of nodes proves to be robust in practice and to produce highly compressed and fast networks. In the future, it will be worth exploring the Shapley value for unstructured pruning and also include other metrics as characteristic functions.

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
