# OpenReview forum: "Shapley Oracle Pruning for Convolutional Neural Networks"
_TMLR — Rejected by TMLR_

### Review · Reviewer_Awja · 2022-09-27

**Summary Of Contributions:**

In this paper, the authors proposed Shapley Oracle Pruning, which uses the Shapley value to measure the average contribution of a neuron. It also incorporates oracle pruning (i.e., the optimal pruning configuration) to build a ranking of nodes to satisfy given constraints. Experiments show that the proposed method achieves decent compression results.

**Requested Changes:**

- Firstly, please differentiate from previous work that already used Shapley value to measure neuron importance (e.g., [4, 12]) and clarify the technical novelty.
- Provide stronger experimental results to show the proposed method can outperform existing work.
- Correct typos and improve writing.

**Strengths And Weaknesses:**

Strengths:
- Firstly, the paper is generally clearly written and easy to follow.
- Ablation studies show the approximation quality of different methods when the real Shapley value is hard to compute.

Weakness:
- Firstly, the proposed method has limited technical novelty. A similar idea on using the Shapley value to measure the importance of each neuron has been proposed in the previous work Neuron Shapley [12] (it was published two years ago in NeurIPS 2020 and cannot be regarded as a concurrent work). It seems that this work applies a similar idea to model compression application, which has limited technical novelty (actually, [4] already applies it to pruning).
- Secondly, the experimental results are not very strong. For ResNet-50 on ImageNet (larger datasets provide a more convincing and realistic comparison), SFP achieves 2% higher accuracy with a comparable FLOPs reduction ratio. The proposed method does not seem to outperform existing work.
- There are some typos. For example:
on page 9, MNIST trained on the LeNet-5, should be LeNet-5 trained on the MNIST dataset;
on page 10, "we are still able apply" -> "we are still able to apply"

---

> ### Author Response · Authors · 2022-10-12
> **Reviewer Awja (Rev. 3) Response**
>
> We attempted to address the reviewer's concerns above, in particular in the Main Response sections Contributions and Numerical Results, and Related Work below.
>
> ### Related work
>
> Here we further address the related methods and differentiate the proposed method. This method has been in effect developed independently from other methods, which, nevertheless, we need to acknowledge and describe in detail.
>
> The proposed work, although similar in a sense of applying the Shapley value to CNNs, differs from [12] in three main points. 1. The idea of the proposed work is aligned with the Efficient AI goals, that is to produce smaller and faster networks. On the other hand, [12] underscores interpretability, provides visualizations, and its primary purpose is to fit into the explainable AI subfield. Moreover, [12] applications are, among all, fairness assessment and finding filters vulnerable to adversaries, rather than network compression. 2. While [12] mentions pruning, it lacks a thorough experimental assessment across several architectures and datasets, as given in the proposed work. 3. Algorithmically, [12] uses a multi-arm bandit algorithm for computing the Shapley value, while we present a set of three different approximations of the Shapley value and compare them against each other for better insight into how to compute the Shapley value for the network compression application.
>
> In this work, we follow a three-step pruning procedure which consists of pre-training, pruning, and re-training. On the other hand, [4] uses a different pruning paradigm, which is iterative pruning, where pruning is done gradually over the course of the training. [4, 29] have also scarce experimental results, with benchmark methods limited to a few basic methods, and not considering larger Resnet networks (or convolutional neural networks, overall, in the case of [29]). This work presents a more thorough assessment which we believe builds a better case to consider the Shapley value when pruning a network.

---

### Review · Reviewer_Q7od · 2022-09-29

**Summary Of Contributions:**

This paper proposes to capture a contribution of a neuron in a neural network with regard to its collaborative effect with other neurons, and then use the calculated contributions of neurons to prune less meaningful ones. To calculate each neuron's contribution score, motivated by the game-theoretical approach, the authors use the Shapley value. Also, since calculating the Shapley value is computationally expensive, the authors further provide three approximation schemes. The results show that the presented Shapley value approximation schemes capture the informativeness of neurons in the neural network, but also the pruned neural network from the approximated Shapley values shows superior performances and efficiencies, compared against the models pruned from relevant pruning baselines.

**Broader Impact Concerns:**

While there are no explicit sections for a broader impact statement, I do not have concerns about the ethical implications of this work.

**Requested Changes:**

---

### Suggestions
**that critically affect my recommendation**
* Based on the first weakness point in the above, the authors should clarify which mechanisms of computing Shapley values, argued as contributions of this paper, are different from existing works.
* There are many unclear sentences, which should be fixed. To mention a few, "the removal of certain nodes lead to a structured mis-classification of inputs": what is the structured mis-classification?; and "for comparison introduce, so called optimal ranking": what is the meaning of comparison introduce and its relation to optimal ranking? I suggest authors to carefully check and revise any unclear sentences.
* The authors could compare the efficiency of the proposed model pruning based on Shapley values against other baselines.

---

### Suggestions
**that would strengthen the work in my point of view**
* The title of Section 5.1 (Ablation study) is not matched with its content. Specifically, the results of Table 2 in Section 5.1 are not ablating some ingredients of the proposed method, but rather comparing its different variants.
* It would be worthwhile to merge results in Table 3, 4, 5, and 6 of varying backbone architectures, if possible.
* The sentence "The size of VGG-16 is reduced to 3.4MB and Resnet to 0.35MB" is not matched with the reported results in Table.

---

### Questions
**that should be also additionally clarified in the paper**
* In Table 2, why the result of Oracle is not 1.0? For my understanding, since $\mathcal{K}_O$ is an oracle subset and $\mathcal{K}_R$ becomes the oracle subset for the Oracle model, I guess the result of Oracle should be 1.0.

**Strengths And Weaknesses:**

---

### Strengths
* The idea of using Shapley value for measuring the importance of a neuron with its relationships to others is interesting, and convincing for neural network pruning mechanisms.
* I like Figure 1, which nicely illustrates the core algorithm of computing Shapley values.

---

### Weaknesses
Regarding the weaknesses, please see the suggestions in the section of Requested Changes below as well.
* The innovations that this paper proposes are unclear, when comparing to the previous work:
- 1. The idea of Shapley value for network pruning is already proposed in (29). Is this method a direct adaption of (29), working on multi-layer perceptrons, to convolutional neural networks?
- 2. The suggested three approximation schemes for computing Shapley values are already suggested in previous methods (1; 6; 4; 21; 2; 12). I understand that the target problem is different (i.e., this paper deals with pruning, whereas, previous works are not), however, beside the problem-side, is there difference between the presented approximation schemes and the ones used before, in the algorithm-side?
* Some sentences are often difficult to understand, because of clarity issues as well as grammar errors.
* The performances and computational efficiencies of the pruned networks from Shapely values are marginal. Specifically, in Table 6, the SFP model proposed in 2018 is not only more effective but also more efficient, compared to the proposed method.
* The pruning from Shapley values seems slow, even if the authors use the approximation scheme. The efficiency comparisons between the proposed and existing methods are required.

---

> ### Author Response · Authors · 2022-10-12
> **Reviewer Q7od (Rev. 2) Response**
>
> We attempted to address the reviewer's concerns in the sections Contributions and Numerical Results in the Main Response, in addition to the sections Approximations and Oracle Ranking below, and Related Work in the Rev. 3 Response.
>
> ### Approximations
>
> The Shapley value is a concept that is computationally hard and previous work on the Shapley value necessarily includes ways to approximate the computation. However, as the reviewers stated, the given approximations have not been properly evaluated for network pruning. This work aims to collect and formulate the approximations to be applicable for pruning internal network units, and compare them for the compression task.
>
> Among the mentioned algorithms, sampling via permutation averaging has been most commonly used for the Shapley value (and one which we found in previous work for network pruning [4]). Kernel Shapley has been motivated by [8] which we adapt and reformulate in Sec. 3.2. The "leave-one-out" pruning has been mentioned in literature [48], but not in relation to the Shapley value. And we have not encountered in literature the generalization for $k>1$, coined here as "partial-k Shapley".
>
> [4] Shapley value as principled metric for structured
> network pruning.
> [8] Extremal principle solutions of games in characteristic function form: Core, chebychev and shapley value generalizations.
> [48] Pruning convolutional neural networks for resource efficient transfer learning.
>
> ### Oracle ranking
>
> Why the result of Oracle in Table 2 is not 1.0?
>
> The column "Oracle" in Table 2 refers to the oracle ranking. Oracle ranking is based on the oracle sets, but the two concepts are different. In short, the accuracy is not 1.0 because we may not be able to build a ranking that includes all the nodes in the oracle sets.
>
> For example, if oracle set for $k=3$ is $\{2,3,6\}$ and for $k=2$ is $\{5,2\}$, then we cannot make a fixed ranking where top-$k$ positions will be equal to the oracle sets (we have four nodes in total but only three top-$3$ spots). Oracle ranking is proposed to maximize the intersection between the elements selected by the top positions in the oracle ranking and the oracle sets.
>
> We attempt to explain these concepts in the second paragraph in the Sec. 5.1 but will provide further explanations to avoid confusion. The details of the oracle ranking can be found in the appendix.
>
> ### Others
>
> Please note that in Table 6, a higher fraction describes a higher compression rate, so the proposed method is a little more efficient (removes 59% of FLOPs vs. 58% removed by SFP).
>
> Thank you very much for pointing to unclear sentences which we indeed missed. The optimal ranking is the oracle ranking. We shall correct those phrases.

---

> > ### Comment · Reviewer_Q7od · 2022-10-26
> > **Thank you for your response.**
> >
> > **Approximations.** Thank you for providing explanations on previous works. As described in my comment to your main response, it is still unclear whether the contribution of Shapley value approximation is direct adaptation of existing works, or there are some improvements which we can consider as contributions of this work. Also, I suggest authors to discuss such points in the paper more concretely as well.
> >
> > **Oracle ranking.** Thank you for providing detailed responses to clarify my question. I suggest authors to include the provided example as well, which will be helpful to understand the Oracle ranking and Oracle sets.
> >
> > **Others.** The authors do not provide responses to my minor comments, listed in my initial review "Suggestions that would strengthen the work in my point of view".

---

### Review · Reviewer_itQS · 2022-09-30

**Summary Of Contributions:**

The paper proposes a pruning model by using the Shapley value. Specifically, it first formalize the pruning as a coalition game where every neuron is a node/player and the reward is the model performance (validation accuracy). Then the Shapley value that is designed for the game could be naturally applied in the network pruning. However, as the it takes the exponential time to calculate the exact  Shaley value, the paper proposes to restrict the permutation space and also use the binary vector to further reduce the cost. The experiments are conduct first to test whether the approximation could still generate a good rank. Then several experiments have been conduct on different architecture and datasets to show the proposed method could achieve a higher compression rate while maintain a good model performance.

**Broader Impact Concerns:**

No ethical concerns as far as I could tell.

**Requested Changes:**

Please refer to the cons.
1. Add a detailed and thorough overview in the related work section on the network pruning.
2. Discuss some potential cases rather than pruning from a pretrained model.
3. Better to add the comparison on the pruning computation budget as well.
4. Add experiments on more compression ratios.

**Strengths And Weaknesses:**

Pros:
1. The paper is well-written and easy to follow.
1. The experiment results look promising.

Cons:
1. The reference section about the network pruning is kind of outdated and a lot of important works are missing such as lottery ticket hypothesis, a thorough review on structural and unstructural pruning etc.
2. All experiments are done on the pruning with a pretrained model. It lacks discussion on its performance on pruning in the middle of training procedure or what the performance would be if the pruned model is retrained.
3. The experiments are done in a little unfair manner. It seems the computation needed for pruning is not being considered. It would be unfair since the pruning by gradient calculation only may need one or two forward pass however the proposed methods need significantly more passes.
4. The experiments are only consider one compression ratio. More ratios are needed to verity the effectiveness.

Minor:
1. There is already several papers [1] and others in related work using Shapley value to conduct pruning. Please have comments on them.

[1] Ancona, Marco, Cengiz Öztireli, and Markus Gross. "Shapley value as principled metric for structured network pruning." arXiv preprint arXiv:2006.01795 (2020).

---

> ### Author Response · Authors · 2022-10-12
> **Reviewer itQS (Rev. 1) Response**
>
> We attempted to address the reviewer's concerns in the Main Response Numerical Results section, and below in the sections Complexity and Related Work. Please also have a look at the Related Work section in the Rev. 3 Response.
>
> ### Pruning complexity
>
> The costs of pruning can be of two sorts, the costs of the pruning procedure and the costs of running the resulting model. The results presented in Tables 3-6 describe the inference computational costs of final pruned models for all the methods.
>
> The computational complexity of network pruning algorithms varies vastly. The common baselines which include the magnitude and derivative pruning require little computation to prune parameters. The gradient methods based on Hessian require just a forward pass to compute gradients (in the case of iterative pruning, a forward pass is made for every iteration). However, these methods may be less effective than other more recent ones. HRank [37] uses the same 500 samples for all of the layers. And for each layer, they need to pass the 500 samples through the network to calculate the feature map. [67] uses over ten thousand samples.
>
> Aware of the computational burden of pruning and to make pruning more practical, we attempt to address the issue by introducing and comparing the Shapley value approximation algorithms. In the paper, we try to explain the complexity analysis of the pruning procedure after introducing each of the approximations. But here we present a more thorough analysis of the computational cost of the method.
>
> In the case of partial-$k$ Shapley value, the cost (the number of samples) is $O(n^k)$ where n is the number of input parameters and $k$ is the size of a subset. This method may require many samples when $n$ and $k$ are larger. Therefore, we suggest subsequent sampling methods. Permutation averaging allows us to control better for the number of samples and in practice, we use 50 samples per layer.
>
> In the case of kernel Shapley, let $s$ be the number of samples required to compute the Shapley value, and $f$ the cost of the forward passes for the validation set. Then, the sampling cost is $sf$. Subsequently, the computation of the Shapley values requires solving linear regression. The complexity of this operation is $O(s^2p+p^3)$, where $s$ is the number of observations/samples and $p$ is the number of features.  Additional cost comes from sorting the values, which is $O(p\log p)$, and is negligible.
>
>
> In sum, the proposed method requires relatively few samples compared to more recent methods but is more demanding than the more traditional benchmarks. Still, sampling could be too costly to perform it at every iteration, and therefore, we perform one-shot pruning. That being said, one should consider the application to select an appropriate pruning method. Fast pruning times would be important for applications where real-time pruning is required. On the other hand, when pruning is performed only once (as it is in the case of the proposed method), and the deployed model is used for further application, the pruning costs may be ignored to large extent. We will further elaborate on the complexity analysis and this discussion in the paper.
>
>
> [37]  Hrank: Filter pruning using high-rank feature map.
> [67] Network trimming: A data-driven neuron pruning approach towards efficient deep architectures
>
> ### Related work
>
>
> The experimental results for this method have been designed with structured pruning in mind due to its practical compression benefits. As a result, we focus on the literature and benchmarks which also deal with structured pruning. However, in a similar manner to the magnitude or derivative-based pruning, the Shapley value could also be applied to unstructured pruning (which we consider as future work). Similarly, the lottery hypothesis has been primarily developed for unstructured pruning. Nevertheless, we shall include a discussion of this line of work in the paper.

---

### Author Response · Authors · 2022-10-12
**Main response**


We would like to thank the reviewers for the assessment of the paper with pointing to the potential weaknesses, which we will try to address below.

At the same time, we are glad that the reviewers find the idea of applying the Shapley value "interesting, and convincing for neural network pruning mechanisms". The paper is considered "well-written" and "easy to follow", and the experiment results "promising". We hope that this work with further improvements can make its way to the TMLR.

### Contributions

To summarize the contributions of this work compared to the existing methods.

The work connects the existing concepts in neural network pruning and develops them into a more general Shapley value framework. For example, leave-one-out pruning is seen as a special case of partial-$k$ Shapley value, oracle rank is proposed to contrast it with the shapley rank as both are derived from the pruning oracle sets.

To alleviate the computational costs of pruning, this work proposes a set of approximations (instead of focusing on only one approximation as given in the previous publications) and, as the only work, performs the analysis in terms of cost-benefit utility for the neural network compression.

To gauge the utility of the Shapley value for the convolutional neural network compression, this work features a variety of CNNs, including modern network architectures (such as Resnets) and datasets (such as Imagenet) to show the broad effectiveness of the Shapley value concept for the particular purpose of finding more efficient neural network models.

### Numerical results

As requested by the reviewers, we are adding more compression ratios for better assessment of the method. Given more time, we will further expand the results. For Resnet-50, please note that the higher number indicated higher compression rate (and a more efficient model).

#### VGG

| *Method*    | *Error* | *FLOPs*  | *Params* |
|-----------|-------|--------|--------|
| **Shapley**   | **7.91**  | 43.0M  | **0.68M**  |
| Dirichlet | 8.48  | **38.0M**  | 0.84M  |
| **Shapley**   | **5.67**  | **109M**   | **3.7M**   |
| SSS       | 6.98  | 183.1M | 3.93M  |
| VP        | 5.72  | 190M   | 3.92M  |


#### Lenet

| *Method*    | *Error* | *FLOPs*  | *Params* |
|-------------|----------|----------|--------|
| **Shapley** | **0.88** | **205K** | 5.9K       |
| SBP         | 0.9      | 226K     | -      |
| **Shapley** | **1.04**| **131K** | **4.5K**|
| Dirichlet   | 1.1      | 140K     | 5.5K   |
| Baseline    | 0.73     | 2549K    | 688K   |


#### Resnet-50

| *Method*    | *Error* | *FLOPs rate*  | *Params rate* |
|-------------|-----------|----------|----------|
| **Shapley** | 24.1      | **55.2** | **71.4** |
| ResRep      | **23.85** | 54.54    | 63.8        |
| **Shapley** | 27.4      | **59.1** | **65.4** |
| SFP         | **25.39** | 58.2     | 57.43        |
| Baseline    | 23.87     | 0.0      | 0.0      |

---

> ### Comment · Reviewer_Q7od · 2022-10-26
> **Thank you for your comments, but the contributions are still unclear**
>
> Thank you for your responses. However, some concerns on contributions in my initial reviews still remain.
>
> Q1. The concept of Shapley value for network pruning is already proposed in (29), and the contribution of this work is adapting the method in (29) targeted MLPs to CNNs. The contribution seems mild, but is clear now.
>
> Q2. The approximation schemes for computing Shapley values are already proposed in previous methods (1; 6; 4; 21; 2; 12). Therefore, in this work, I am still wondering the contribution is to compare multiple approximation algorithms and then report the best result, or improving the existing approximation algorithms. If this is the first case, the contribution seems mild.

---

### Decision · Action_Editors · 2022-11-16

**Recommendation:** Reject

**Comment:**

The paper proposes a new pruning model with Shapley value. All the reviewers give negative decision recommendations.The reviewers have concerns about the contribution and experimental results. As mentioned by Q7od, the idea of Shapley value for network pruning is proposed by the existing methods. Though the authors further explain the contribution, the difference seems still margin. The reviewers also point that the proposed method does not suppress SOTA pruning methods in the experimental results. More emperimental analyses are required as well. For the above reasons, the paper is not ready to be published on TMLR.

**Audience:**

Though pruning with Shapley value is reasonable, this paper brings limited new knowledge compared with existing methods. It is better to revise this paper to enrich the contribution.

**Claims And Evidence:**

The experimental analyses can partly support the effectiveness of pruning with Shapley value. However, more experiments are required to verify its superiority to the existing methods.